# Data Driven Water Surface Elevation Forecasting Model with Hybrid Activation Function—A Case Study for Hangang River, South Korea

**Hyung Ju Yoo [1], Dong Hyun Kim [1], Hyun-Han Kwon [2] and Seung Oh Lee [1],***

[1] Department of Civil Engineering, Hongik University, Seoul 05066, Korea; hyungzu11@gmail.com (H.J.Y.); uou543@gmail.com (D.H.K.)
[2] Department of Civil and Environmental Engineering, Sejong University, Seoul 05006, Korea; hkwon@sejong.ac.kr
*    Correspondence: seungoh.lee@hongik.ac.kr

**Abstract:** To date, physical, numerical or data-driven models have been used to forecast water surface elevation in rivers for specific times or locations in the literature. Recently, the trend of forecasting water surface elevation changed from physical and numerical models to data-driven models with the help of the development of big data processing technology and fast simulating time of data-driven models. In this study, a data-driven model with Long Short-Term Memory (LSTM) was developed using TensorFlow, one of the famous deep learning frameworks and forecasting of water surface elevation affected tidal river was performed in Hangang River, Korea. From many types of field measurements, the hourly hydrological data, precipitation, outlet discharge of dam upstream and tidal levels were selected as the input dataset through a *t*-test and a *p*-value. In particular, the hybrid activation function was proposed to alleviate the vanishing gradient and dying neuron problems generally issued in the application of the activation function. The model showed that the root mean square error (RMSE) and peak error (PE) decreased by 0.22–0.25 m and 0.11–0.21 m, respectively, and the Nash-Sutcliffe efficiency (NSE) increased up to 79.3%–97.0% compared with the single activation functions. For $w_1 = 0.6$ and $w_2 = 0.4$ in the hybrid activation function, the improvement of accuracy and the enhancement of the application range of the leading time interval were obtained through a sensitivity analysis. Moreover, the hybrid activation function showed a good performance. The forecasting results provided by this model can be used as reference data for the establishment of the emergency action plan (EAP).

**Keywords:** flood forecasting; Long Short-Term Memory (LSTM); hybrid activation function; Hangang River

## 1. Introduction

Recently, flood damage has increased at riversides because the water surface elevation is rising rapidly as the occurrence of extreme flooding has increased due to global warming and climate change [1,2]. As an example of flood damage to riversides, flooding occurred in roads, parks and parking lots and caused damage to life and property [3]. Therefore, it was necessary to immediately forecast and alert about the possibility of the occurrence of flood damage on riversides caused by flash floods.

For forecasting flooding that occurs along a riverside, many countries have provided flood forecasting system services such as the Advanced Hydrologic Prediction Service (AHPS in United States), and the Evaluation et Suivi des Pluies en Agglomération pour Devancer l'Alerte (ESPADA

in France) by using hydrological data [4]. In general, the precipitation data have been used for a precipitation-runoff model and the water surface elevation in rivers has been forecasted sequentially. As the final step, the water surface elevation of the river has been used to determine whether flood damage occurs at riverside [5]. This is a standardized flood forecasting process around the world, including Korea. Therefore, accurate forecasting of the water surface elevation has been most significant to forecast a flood event. There are typically two methods for forecasting the water surface elevation of a river. The first one is based on a numerical model which can analyze various differential equations such as Navier–Stokes equations and calculates the water surface elevation by using geometry data of a river, discharge and precipitation, etc. However, a numerical analysis by computational fluid dynamics (CFD) has provided accurate and sophisticated forecasting results, whereas it is difficult to acquire related essential data such as geometry data before a simulation and a simulation takes more time when 2D or 3D results are expected. Therefore, the numerical model had a limitation in forecasting and warning floods within limited time [6]. Recently, the forecasting of water surface elevation was performed by using a data-driven model based on the statistical relationship between the input data and the output result as a large amount of data has been collected and big data processing technology has been developed. Examples of data-driven models applied in the forecasting of water surface elevation are statistical models which include linear regression, autoregressive moving average (ARMA) and autoregressive integrated moving average (ARIMA) models and machine learning (ML) models such as artificial neural networks (ANN) [7]. In addition, the ML models were good at performance for forecasting time series data such as water surface elevation compared with other models [8]. In addition, the data-driven model using the artificial neural network has the advantage because it is easy to acquire data when compared to physical models and the time required for forecasting water surface elevation was short, although the ANN had problems such as dependence on the state of the input data, unexplained behavior of the network, etc. Therefore, the data-driven model using ANN was developed to forecast the water surface elevation in this study. Various artificial neural network models were applied for forecasting the water surface elevation. The long short-term memory (LSTM) showed a good performance for forecasting the water surface elevation. However, the LSTM tends to under-forecast at a high water-surface elevation and has a limitation in that the accuracy of forecasting decreases as the leading time becomes longer. Nevertheless, research which solved these problems have been insufficient. In general, the activation functions used in LSTM include the sigmoid function, the hyperbolic tangent function and the rectified linear unit function, but each activation function causes problems such as a vanishing gradient and a dying neuron [9–11]. Therefore, in this study, a hybrid activation function in the form of a combination of the hyperbolic tangent function and the rectified linear unit function with a weighting factor was proposed to improve the forecasting accuracy at a high water surface elevation and for a long leading forecasting time. The LSTM in Tensorflow, a deep-learning open-source software library provided by Google, was applied and the hourly hydrological data from 2009 to 2018 were used to forecast the water surface elevation in Hangang River, Korea. Finally, the hybrid activation function was applied to evaluate whether the accuracy of forecasting improved at a high water surface elevation the range of leading time was reviewed for highly accurate forecasting.

## 2. Literature Review

### 2.1. Numerical Model

Various studies were performed to forecast the water surface elevation using numerical model based on physical laws. Rinaldi et al. [12] applied the flood event and forecasted the water surface elevation of the Cecina river to simulate the levee breach by using the Delft 3D model. Teng et al. [13] introduced various one-, two- and three-dimensional numerical models for flood modeling through forecasting water surface elevation and compared the advantages and disadvantages of each model. In Korea, Lee and Lee [14] reviewed the change of water surface elevation in the Hangang River due to

change of tidal level and Paldang dam outlet discharge by using FLDWAV, which is a one-dimensional unsteady flow model. Song et al. [15] developed a numerical model that discretized the shallow water equation to analyze the backwater effect according to the discharge and forecasted the change of the water surface elevation in the river. However, the forecasting water surface elevation using these numerical models had some limitations, such as difficulty in acquiring geometry data and the longer simulation time. Thus, as an alternative approach, various artificial neural network models were used to forecast the water surface elevation.

### 2.2. Artificial Neural Network Model

Recently, forecasting the water surface elevation using the artificial neural network models was attempted to solve the problems in a physical or a numerical model. For instant, Yeo et al. [16] performed a short-term forecasting of the water surface elevation using the ANN model for the water surface elevation station in the Gamcheon Basin. Chen et al. [17] constructed an ANN-based forecasting model in the river and reviewed the applicability of the model. Hidayat et al. [18] verified the accuracy of the forecasting model by constructing an ANN model for the Mankam River in Indonesia and used tributary water surface elevation and tidal level to construct a real-time forecasting system. However, the ANN model did not consider the past data in learning the data. Thus, the recent research trend about forecasting water surface elevation using artificial neural network changed from ANN, recurrent neural network (RNN) models to long short-term memory (LSTM). Coulibaly and Anctil [19] confirmed that the RNN model derived effective real-time forecasting results by applying the RNN model to forecast the short-term discharge and reservoir level in Gondo aquifer, Burkina Faso. Supharatid [20] used the Multi-Layer Neural Network (MNN) to forecast the tidal level in the Chao Phraya River estuary, Thailand, and generated a relationship curve between water surface elevation and discharge in the tidal stream by using the tidal level. Yoo et al. [21] forecasted the water surface elevation in Hangang River and compared the forecasting accuracy of each model by applying ANN, RNN and a nonlinear autoregressive network with the exogenous (NARX) model. Thus, they confirmed that the NARX neural network model was most suitable for forecasting water surface elevation by analyzing the error of the forecasting result. Zhang et al. [22] confirmed that the LSTM is superior to the previous feed-forward neural network (FFNN) model in predicting water surface elevation over a long-term period. Tran and Song [23] used the RNN, RNN-BPTT, and LSTM to perform water surface elevation forecasting of the Trinity River in Texas, USA, and showed that the LSTM had a good performance for forecasting the water surface elevation. Jung et al. [24] used the LSTM to forecast the upstream water surface elevation in the Geumgang river basin, Korea. The accurate forecasting was performed for the entire water surface elevation. However, the forecasting result was underestimated at a high water surface elevation. Jung et al. [25] forecasted the water surface elevation of the Jamsu bridge in Hangang River, Korea, using the LSTM and confirmed that the forecasting accuracy of the model decreased as the forecasting leading time was longer. Through the sound results from the research mentioned above, the LSTM was chosen in this study. Furthermore, the hybrid activation function was proposed to resolve the underestimation at a high water surface elevation and improve the accuracy of forecasting in longer leading times.

## 3. Methodology

### 3.1. Long Short-Term Memory Model

The LSTM was proposed by Hochreiter and Schmidhuber to solve the problems of optimization hurdle and vanishing gradient in the RNN model. The problems of the optimization hurdle and the vanishing gradient were a difficulty found in training artificial neural networks with gradient-based learning methods and backpropagation. The gradient is vanishingly small, effectively preventing the weight from changing its value. Moreover, these problems completely stop the neural network from training. To make matters worse, this occurred frequently, especially in long-time series data.

Therefore, the LSTM could be applied to forecast long-time series data because of its efficiency in identifying long-term dependence over time and this has been studied in various fields [26,27].

The main elements of LSTM are memory-moving cells, which can maintain the state over time and three gates that control the transfer of data in and out of cells, unlike the RNN model [28]. The LSTM applies the concept of a cell to update the state of a specific time ($h_t$) and determines whether to update the internal information or not by using the input data and the state so far. The types of gates for controlling data transfer of the cell are: a forget gate ($f_t$), an input gate ($i_t$) and an output gate ($o_t$).

First the forget gate ($f_t$) applies the output of the previous cell ($h_{t-1}$) and the current input data ($x_t$) to the sigmoid activation function to obtain a value between 0 and 1 and determines whether to maintain or remove the input information. This can be shown in Equation (1):

$$f_t = \sigma\big(W_f \cdot [h_{t-1}, x_t] + b_f\big) \tag{1}$$

where σ is the sigmoid activation function, $W_f$ is the weight of the forget gate, and $b_f$ is the bias of the forget gate.

The second the input gate ($i_t$) decides which input information is stored in the cell and which information is updated by using the sigmoid activation function. The input gate also generates a candidate cell ($N_t$) which is used when updating a new cell state by using the hyperbolic tangent activation function. The candidate cell is expressed through the following equations:

$$i_t = \sigma(W_i \cdot [h_{t-1}, x_t] + b_i) \tag{2}$$

$$N_t = tanh(W_n \cdot [h_{t-1}, x_t] + b_n) \tag{3}$$

where $W_i$, and $W_n$ denote the weights of the input gate and the candidate cell, respectively, $b_i$, $b_n$ denote the bias of the input gate and the candidate cell, respectively.

Next, the current cell state ($C_t$) is updated by combining the previous cell state ($C_{t-1}$) and the candidate cell ($N_t$), as shown in Equation (4):

$$C_t = f_t \cdot C_{t-1} + i_t \cdot N_t \tag{4}$$

The output gate ($o_t$) determines which part of the cell state is output by using the sigmoid activation function, as shown in Equation (5) below. Finally, the hyperbolic tangent activation function is used to update the state of a specific time ($h_t$) by multiplying it with the activated cell state ($C_t$).

$$o_t = \sigma(W_o \cdot [h_{t-1}, x_t] + b_o) \tag{5}$$

$$h_t = o_t \cdot \tan h(C_t) \tag{6}$$

where $W_o$ is the weight of the output gate and $b_o$ is the bias of the output gate.

Previous studies have shown that the LSTM could be used to accurately forecast water surface elevation [25]. However, the water surface elevation is under-forecasted at the high water surface elevation conditions because the hyperbolic tangent activation function (value range from −1 to 1) was basically used when multiplying the output gate and the cell state in the structure of LSTM. Thus, the hyperbolic tangent activation function should be changed another activation function to solve under-forecasting at the high water surface elevation conditions. In general, activation functions include linear and nonlinear functions. The linear function changes by a constant multiple of the input and the form of the linear function is a straight line. However, the use of a linear function as an activation function is not an advantage of the neural network. On the other hand, the non-linear functions have two or more straight or curved forms and the typical nonlinear functions used in neural network are the sigmoid function, the hyperbolic tangent function, the rectified linear unit (ReLU) function, etc. However, the use of an existing activation function, such as the hyperbolic tangent function and the ReLU function, also causes vanishing gradient problems and dying neuron problems

in error back propagation, leading to under-forecasting or over-forecasting results [9–11]. Therefore, the activation function of the LSTM needed to be changed from the hyperbolic tangent function to a new type of activation function to accurately forecast the water surface elevation in high water surface elevation conditions.

*3.2. Hybrid Activation Function*

The activation function is used to determine the activation and deactivation of output data when a signal of input data is received in the neural network structure. It is important to select an activation function that is suitable for the purpose of the study because the results are dependent on which activation function is used. In general, activation functions include linear and nonlinear functions. The linear function changes by a constant multiple of the input and the form of the linear function is a straight line. However, the use of the linear function as an activation function is not an advantage of the neural network. On the other hand, the non-linear functions have two or more straight or curved forms and typical nonlinear functions used in neural network are the sigmoid function, the hyperbolic tangent function, the rectified linear unit (ReLU) function, etc. The sigmoid function has the feature of a fast learning speed. In addition, it has a value of 0 to 1 in the range of $-\infty$ to $\infty$. The form of the sigmoid function is shown in Equation (7):

$$\sigma(x) = \frac{1}{1 + e^{-x}} \tag{7}$$

$$\sigma'(x) = \sigma(x)(1 - \sigma(x)) \tag{8}$$

However, $x$ has a value close to 0 in the range $-\infty$ to $\infty$ in the derivative form of the sigmoid function and it causes vanishing gradient problems in error back propagation [29].

The hyperbolic tangent function can be expressed as the ratio of the hyperbolic sine function and the hyperbolic cosine function or as the ratio of the sum and difference of the two exponential functions. The form of the hyperbolic tangent function is shown in Equation (9):

$$tanh(x) = \frac{sinh(x)}{cosh(x)} = \frac{e^x - e^{-x}}{e^x + e^{-x}} \tag{9}$$

$$tanh'(x) = 1 - tanh^2(x) \tag{10}$$

The hyperbolic tangent function is like the sigmoid function, it has a value of $-1$ to 1 in the range $-\infty$ to $\infty$. However, the hyperbolic tangent function also has a value close to 0 when $x$ is $-\infty$ and $\infty$ in the derivative form which causes a problem of vanishing gradient during back propagation [30].

The rectified linear unit (ReLU) function was first introduced by Hahnloser et al. [28] and was used as an activation function in neural network model. The ReLU function changes its form of function, depending on the range of $x$. The ReLU function is shown in Equation (11):

$$f(x) = \begin{cases} 0 & if \ x < 0 \\ x & if \ x \geq 0 \end{cases} \tag{11}$$

$$f'(x) = \begin{cases} 0 & if \ x < 0 \\ 1 & if \ x \geq 0 \end{cases} \tag{12}$$

The ReLU has a value of 0 when $x$ is less than 0, and $f(x) = x$ when $x$ is greater than 0, as shown from the above function equation. However, the ReLU function has a dying neuron problem because $f(x)$ always has a value of zero when the value of $x$ is less than zero. Moreover, it has the limitation of overfitting the results [30].

In this study, the hybrid activation function was proposed to solve the problems of general activation functions such as the vanishing gradient problem, the dying neuron problem, etc. The

hybrid activation function is composed of a hyperbolic tangent function and a rectified linear unit function. The form of the hybrid activation function is as follows:

$$f(x) = w_1 ReLU(x) + w_2 tanh(x) \tag{13}$$

$$f'(x) = \begin{cases} w_1 + w_2\left(1 - tanh^2(x)\right) & if \quad x < 0 \\ w_2\left(1 - tanh^2(x)\right) & if \quad x \geq 0 \end{cases} \tag{14}$$

where $w_1$ and $w_2$ are weights, and each sum of weights is one ($\sum_{i}^{n} w_i = 1.0$). As the weights ($w_1$ and $w_2$) change, the range of the hybrid activation function occupies the hatched part of the figure (see Figure 1).

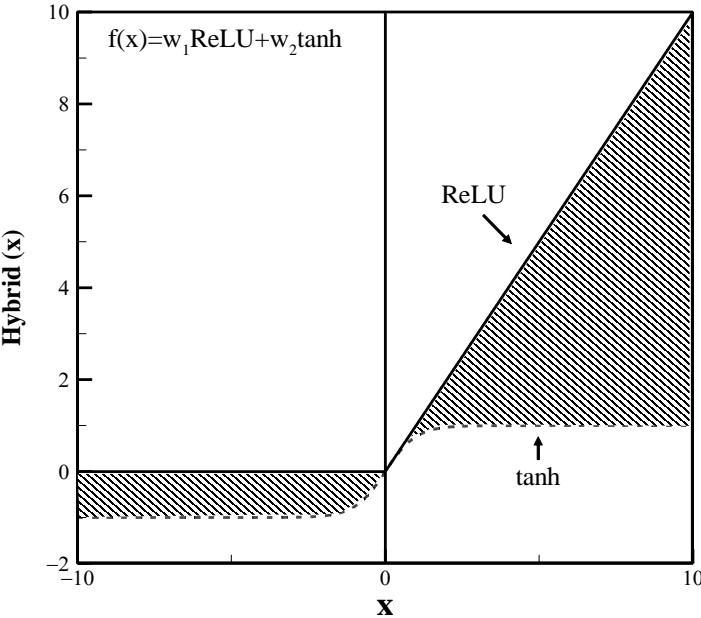

**Figure 1.** Hybrid activation function (Dash line: hyperbolic tangent activation function, Line: ReLU activation function, Hatched Area: hybrid activation function).

### 3.3. Criteria for Comparison of Model Performance

In this study, the forecasting accuracy of model was compared through a statistical analysis of the forecasting results after learning the input data through the flood forecasting model. The root mean square error (RMSE) and Nash-Sutcliffe efficiency (NSE) were used as indexes for scrutinizing the accuracy of the model. Moreover, the peak error (PE) was also used to evaluate the performance of model. The error and coefficient calculation formulas are as follows [31]:

$$RMSE = \sqrt{\frac{\sum_{i=1}^{n}(O_i - P_i)^2}{n}} \tag{15}$$

$$NSE = \left(1 - \frac{\sum_{i=1}^{n}(O_i - P_i)^2}{\sum_{i=1}^{n}\left(O_i - \overline{O_i}\right)^2}\right) \times 100 \tag{16}$$

$$PE = (O_{i,max} - P_i) \tag{17}$$

where $O_i$ and $P_i$ are the observed water surface elevation and the forecasting water surface elevation for time t, respectively, $\overline{O_i}$ is the average value of the observed water surface elevation, $O_{i,max}$ is the

maximum value of the observed water surface elevation, *i* is the *i*-th observed dada or *i*-th forecasting results and n is the total number of observed data.

In the case of RMSE and PE, the closer the value is to zero, the better the forecasting value and the observed value are matched. When the forecasting value and the observed value are well matched, the value of NSE approaches 1.

### 3.4. Autocorrelation Coefficient (AC) and Partial Autocorrelation Coefficient (PAC)

The successive series of observations may be correlated with each other. Thus, the index for determining the linear relationship between the lagged values of the time series includes the autocorrelation coefficient and the partial autocorrelation coefficient. The autocorrelation coefficient is an index of the correlation between the data, $y_t$ and $y_{t+k}$ for a lag time t. The coefficient calculation formula is as follows [32]:

$$r_k = \frac{\sum_{t=1}^{n-k}(y_t - \overline{y})(y_{t+k} - \overline{y})}{\sum_{t=1}^{n}(y_t - \overline{y})^2} \tag{18}$$

where *k* is the number of autocorrelations in the function, *n* is the number of observations in the series and $\overline{y}$ is the mean value.

The partial autocorrelation coefficient is an index of the correlation between data, $y_t$ and $y_{t+k}$ which excludes influence from observations with any lag time other than time *t*. The coefficient calculation formula is as follows:

$$p_{k,k} = \frac{r_k - \sum_{j=1}^{k-1} p_{k-1,j} r_{k-j}}{1 - \sum_{j=1}^{k-1} p_{k-1,j} r_j} \tag{19}$$

where $p_{k,j} = p_{k-1.j} - p_{k,k} p_{k-1,k-j}$, *k* is the number of autocorrelations in the function.

In general, if the autocorrelation coefficients are persistently large. This indicates that the time series is probably non-stationary. Therefore, the concept of a 95% confidence interval which assumes that it is not different from zero within a 95% confidence interval, was used in autocorrelation and partial autocorrelation. The 95% confidence interval is calculated as follows:

$$95\% \ CI = \pm 1.96 \frac{1}{\sqrt{n}} \tag{20}$$

where CI is the confidence interval and the value of 1.96 represents the area under the normal curve.

The autocorrelation coefficient (AC) and partial autocorrelation coefficient (PAC) are widely used in statistics and econometrics, and in time series analysis by using the ARIMA model. For example, there are analyses of the Computer I/O pattern, forecasting of water quality (water temperature and concentration of dissolved oxygen), forecasting of supply and demand about electricity, etc. [33–35].

## 4. Study Area and Data Selection

### 4.1. Study Area

The Hangang River basin is located in the central part of the Korean Peninsula with a north latitude of 36°30′ to 38°55′ and an east longitude of 126°24′ to 129°02′. The basin area is 25,954 km$^2$, and the length of the river is 494 km, accounting for about 23% of South Korea's total land area. In this study, the study section was approximately 91.35 km from Paldang dam to the West sea (see Figure 2). The 12 parks and 27 bridges are in the downstream of Paldang dam. In addition, the Hangang River has no estuary banks, so it is affected by the tide. In addition, forecasting the water surface elevation is difficult because it is a tidal river.

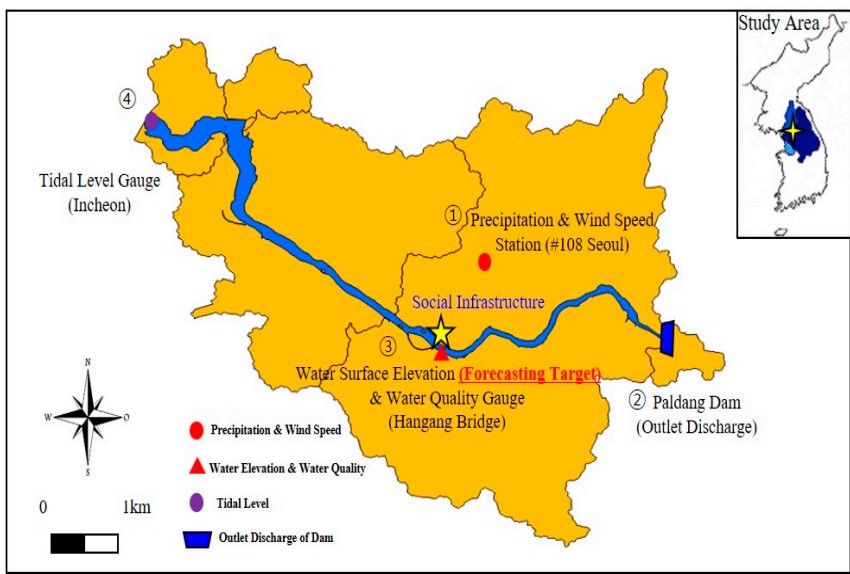

**Figure 2.** Study area (Hangang River).

### 4.2. Input Data

The hybrid activation function was proposed to forecast the water surface elevation, which is an input data with time series characteristics in this study. The purpose of this study was to evaluate the performance of the forecasting model based on the use of the hybrid activation function. Therefore, the type of input data, quantity, and correlation of input data to be used for forecasting the water surface elevation were considered. Yoo et al. [21] have investigated the accuracy of forecasting according to a combination of input datasets in Hangang River. They found that the combination of precipitation, outlet discharge of dam, water surface elevation and tidal level showed a good performance of forecasting water surface elevation in view of forecasting accuracy. Therefore, we also applied the same input data as that previous study to forecast the water surface elevation in Hangang River. Therefore, the input data used for forecasting the water surface elevation are precipitation (① in Figure 2), outlet discharge of Paldang dam (② in Figure 2), the water surface elevation of Hangang River bridge (③ in Figure 2), the tidal level of Incheon (④ in Figure 2), wind speed (① in Figure 2) and the concentration of dissolved oxygen (③ in Figure 2). The observed data for 10 years (from 2009 to 2018) on an hourly basis from the Korea meteorological administration, the Hangang River flood control center, and the Korea hydrographic and oceanographic agency, Water Environment Information System, respectively, were used. The location of the station and the retention periods of the collected data in this study are presented in Table 1 and the units of observed data are as follows. The unit of precipitation is millimeters, the unit of outlet discharge is cubic meters per second, the unit of water surface elevation is meters, the unit of tidal level is centimeters, the unit of wind speed is meters per second, and the concentration of dissolved oxygen is milligrams per liter.

**Table 1.** Information about hydrological stations.

| No | Stations | Items * | Latitude | Longitude | Period |
|----|----------|---------|----------|-----------|--------|
| 1 | Seoul | P | 37°34′ | 126°57′ | |
| 2 | Paldang | O | 37°31′ | 127°17′ | |
| 3 | Hangang Bridge | E | 37°30′ | 126°57′ | 10 years |
| 4 | Incheon | T | 37°27′ | 126°35′ | (2009–2018) |
| 5 | Seoul | W | 37°34′ | 126°57′ | |
| 6 | Hangang Bridge | D | 37°30′ | 126°57′ | |

* P = Precipitation; O = Outlet Discharge; E = Water surface elevation; T = Tidal level; W = Wind speed; D = Concentration of Dissolved Oxygen.

### 4.2.1. Preprocessing of Data

The collected data have a missing value and outliers due to a malfunction of the measuring instruments, crosstalk in the radio wave path and changes in the measurement points. Thus, preprocessing of the input data was required to improve the accuracy of the forecasting model for this study. The outliers and the missing value of observed data were corrected and interpolated according to the article 14 of the installation environment, maintenance, and management of hydrological survey facilities and the quality control standard for hydrological data [36]. The interpolation and correction methods are summarized in Table 2.

**Table 2.** Correction method for outlier and missing value of hydrological data [36].

| Method | Contents |
| --- | --- |
| Use of relevant station data | Correct using a normal value of station<br>Correct with linear interpolation<br>The thresholder decides and correct the value |
| Use of data from nearby stations | Correct using a relationship with upstream and downstream station values |

The proposed method was used to correct and interpolate outliers and missing values corresponding to about 1% of the total data. Most of the outliers and missing values of hydrology data used in this study occurred in the non-flood season which was not considered in this study.

### 4.2.2. *t*-test and *p*-value

The *t*-test analysis was performed by classifying all the input data according to whether flood damage had occurred at the riverside or not and the *p*-value was derived to determine whether the collected data are the main factors for forecasting the water surface elevation. The *t*-test analysis is a statistical technique needed to determine whether the difference in mean value between two groups is statistically significant. The following assumptions must be satisfied in the *t*-test analysis.

Firstly, the data should be continuous numbers with equal intervals (identical interval and continuity). Secondly, the two groups must be independent of each other (independence). Thirdly, the numerical value of the data should be normal (normality).

The *p*-value was derived if the above assumptions were met. The process of derivation is shown in Figure 3. The factor could be considered as a significant factor if the *p*-value is less than the value which the researcher set as the threshold and the value of the threshold was 0.05 in general [37]. The results of the *t*-test analysis and the *p*-value for the various input data, such as precipitation, outlet discharge of Paldang dam, water surface elevation of Hangang River, tidal level of Incheon, wind speed, and concentration of dissolved oxygen are summarized in Table 3. Moreover, precipitation, outlet discharge Paldang dam, water surface elevation of Hangang River, and tidal level of Incheon used as input data in this study. Because these factors were considered to be important factors in forecasting the water surface elevation through the results of the *t*-test analysis and *p*-value. On the other hands, the wind speed (*p*-value = 0.06) and concentration of dissolved oxygen (*p*-value = 0.29) were not important factors for forecasting the water surface elevation.

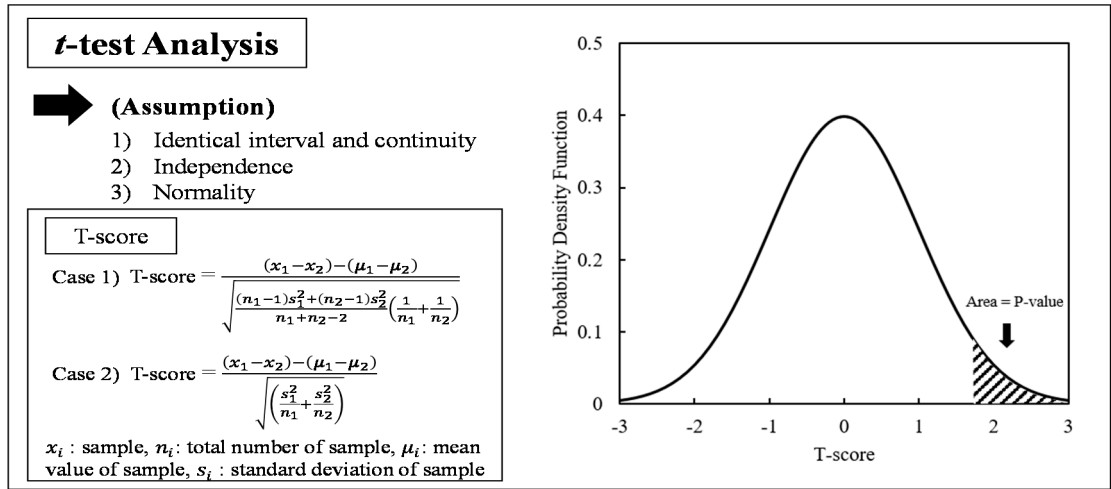

**Figure 3.** *p*-value derivation process.

**Table 3.** *t*-test analysis for input data.

| Input Data | *t*-score | *p*-value | Significant Factor for Forecasting Flood |
|---|---|---|---|
| Precipitation | 8.9 | ~0.00 | O |
| Outlet Discharge | 41.2 | ~0.00 | O |
| Water Elevation | 37.8 | ~0.00 | O |
| Tidal Level | 2.2 | 0.03 | O |
| Wind Speed | 1.9 | 0.06 | X |
| Concentration of Dissolved Oxygen | 1.0 | 0.29 | X |

## 5. Model Setup

The open source library was used in this study. Python (version 3.6.4, Anaconda) was used as the programming language. Numpy (version 1.14.2) and Pandas (version 0.22.0) libraries inside Python were used for data management to execute the forecasting model. The forecasting model using the LSTM was developed by using the Tensorflow (version 1.8.0) provided by Google as an open source library. The TensorFlow is composed of data flow graph structure and is used in various research fields of machine learning and deep learning. Moreover, the forecasting of water elevation was performed by using a computer with the following specifications: Intel Core i5-6600 CPU (Central Processing Unit), 8.00 GB of RAM (Random Access Memory), 256 GB of SSD (Solid State Drive), and an Intel(R) HD Graphics 530 video card.

### 5.1. Sensitivity Analysis

The sensitivity analysis was performed on various factors in the LSTM prior to evaluating the performance of the forecasting model and the performance of the newly proposed hybrid activation function in this study. The simulation cases for the sensitivity analysis are summarized in Table 4. The four parameters can be set in LSTM. The hydrological data were used for 1 year as input data. The forecasting data concern water surface elevation after 3 h because the leading time of the flood forecasting in river is within 3 h. The uncertainty of each parameter was expressed as the standard deviation and the average CPU time was verified with 30 iterative simulations.

**Table 4.** Sensitivity analysis for LSTM for forecasting flood (*: reference value).

| Parameter | Value | Evaluation |
|-----------|-------|------------|
| Hidden Layer (A) | 1 *, 3, 5, 10, 20 | |
| Learning Rate (B) | 0.005, 0.01 *, 0.02, 0.05, 0.1 | Standard deviation |
| Sequence Length (C) | 1 *, 3, 6, 9, 12, 24 | (Uncertainty) and CPU Time |
| Iterations (D) | 1, 10, 100 *, 1000, 10,000 | |

The standard deviation and the CPU time were considered to derive the optimal setting value for each parameter. The standard deviation and CPU time for each parameter are shown in Figure 4 and the results of sensitivity analysis are summarized in Table 5.

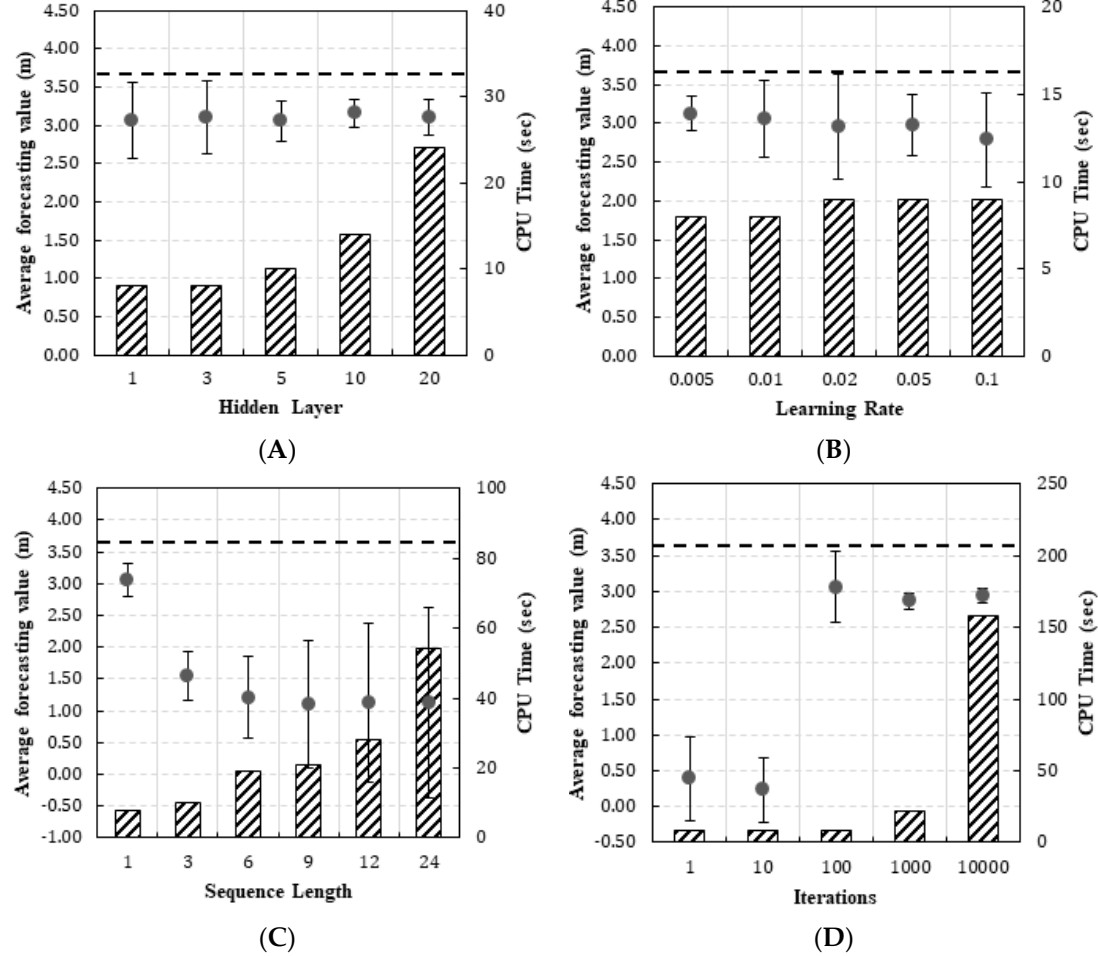

**Figure 4.** Results of the sensitivity analysis ((**A**): hidden layer; (**B**): learning rate; (**C**): sequence length; (**D**): iterations; dashed lines: average value of observed data; ▨ CPU Time, ● Average forecasting value).

**Table 5.** Sensitivity analysis for parameter.

| Parameter | Value | Uncertainty (m) | CPU Time (s) |
|---|---|---|---|
| Hidden Layer | 1 | 0.50 | 8 |
| | 3 | 0.48 | 8 |
| | 5 | 0.27 | 10 |
| | 10 * | 0.18 | 14 |
| | 20 | 0.23 | 24 |
| Learning Rate | 0.005 * | 0.22 | 8 |
| | 0.01 | 0.50 | 8 |
| | 0.02 | 0.67 | 9 |
| | 0.05 | 0.40 | 9 |
| | 0.10 | 0.61 | 9 |
| Sequence Length | 1 * | 0.25 | 8 |
| | 3 | 0.38 | 10 |
| | 6 | 0.65 | 19 |
| | 9 | 1.00 | 21 |
| | 12 | 1.25 | 28 |
| | 24 | 1.50 | 54 |
| Iterations | 1 | 0.58 | 8 |
| | 10 | 0.45 | 8 |
| | 100 | 0.50 | 8 |
| | 1000 * | 0.11 | 22 |
| | 10,000 | 0.10 | 158 |

*: Optimal value for flood forecasting model.

As a result of the sensitivity analysis, the optimal values for each parameter were those with a hidden number of layers of 10, the learning rate was 0.005, there were 1000 iterations, and the sequence length was 1 h in this study. However, in the case of sequence length, the setting value can be changed depending on how the engineer sets the leading time. The recommended time of the sequence length is more than 5 h, according to previous studies [25].

*5.2. Model Scenarios*

The optimal values derived for each parameter of LSTM through the sensitivity analysis were applied in this study. Prior to setting up the model scenarios, the input data used in this simulation were precipitation, outlet discharge of Paldang dam, water surface elevation of Hangang River, and tidal level of Incheon, as mentioned in the previous section, and all the data were obtained from the station at the study area. The collected input data is for 10 years. Moreover, two scenarios were set up to evaluate the performance of the flood forecasting model and to evaluate the performance of the hybrid activation function proposed in this study.

In the first scenario, the weights of the hybrid activation function were set from the near hyperbolic tangent function ($w_1 = 0.1$ and $w_2 = 0.9$) to the near rectified linear unit function ($w_1 = 0.9$ and $w_2 = 0.1$) because, the hyperbolic tangent function has an under-forecasting problem due to vanishing gradient and the rectified linear unit function has an over-forecasting problem due to dying neuron. Therefore, we compensated for these problems by changing the weights and found the optimal value of weights for accurate forecasting. Moreover, the trend in frequency of utilization is moving from hyperbolic tangent function to rectified linear unit function in many researches [30]. Therefore, the forecasting accuracy of the hybrid activation function proposed in this study was checked and the change of the forecasting accuracy was compared by changing the weights ($w_1$ and $w_2$) of the hybrid activation function. The second scenario involved checking the applicability of the maximum forecasting time when using the hybrid activation function and the change in the forecasting accuracy was compared by changing the leading time. The simulation cases were summarized in Table 6.

**Table 6.** Simulation cases for the application of the hybrid activation function.

| Scenario | | | | |
|---|---|---|---|---|
| I | | | II | |
| Case | $w_1$ | $w_2$ | Case | Leading Time Interval |
| 1 | 0.1 | 0.9 | A | 1 h |
| 2 | 0.2 | 0.8 | B | 3 h |
| 3 | 0.3 | 0.7 | C | 6 h |
| 4 | 0.4 | 0.6 | D | 9 h |
| 5 | 0.5 | 0.5 | E | 12 h |
| 6 | 0.6 | 0.4 | F | 15 h |
| 7 | 0.7 | 0.3 | G | 18 h |
| 8 | 0.8 | 0.2 | H | 21 h |
| 9 | 0.9 | 0.1 | I | 24 h |

The 7 years of input data (from 2009 to 2015) were used as the training dataset and the 3 years of input data (from 2016 to 2018) were used for the test dataset in order to simulate the scenarios. In particular, 2016 was selected for model evaluation because the maximum water surface elevation occurred in 2016. Moreover, the observed water surface elevation corresponding to about 50 h from 5–7 July 2016 was compared. The maximum water surface elevation (EL. 5.61 m) occurred on 5 July 2016 at 10 p.m. In addition, the input data were normalized to improve the accuracy of the forecasting model and to improve the learning speed. The minmax normalization was used and rescaling (from 0 to 1) of the input data was applied to the simulation.

## 6. Results and Analysis

The training dataset and the test dataset were applied independently in this study. The performance of the forecasting model for each scenario was evaluated by using model evaluation indexes such as NSE, RMSE, and PE, which were derived by comparing the forecasting result and the observed data. In addition, the forecasting water surface elevation and the observed water surface elevation were compared for about 50 h on 5 July 2016 when the maximum water surface elevation in three years (from 2016 to 2018) occurred.

### 6.1. Results from Scenario I

The result of the improvement of the forecasting accuracy when using the hybrid activation function proposed in this study are summarized in Table 7. In addition, the forecasting result and the observed water surface elevation are presented in Figure 5 according to the change of weight ($w_1$ and $w_2$).

**Table 7.** Summary of accuracy for various weights in Scenario I (Period: 5–7 July 2016, about 50 h).

| Case | RMSE (m) | NSE (%) | PE (m) | Remark |
|---|---|---|---|---|
| I-1 | 0.38 | 98.4 | 0.13 | No good |
| I-2 | 0.47 | 97.6 | 0.28 | No good |
| I-3 | 0.41 | 98.1 | 0.10 | No good |
| I-4 | 0.48 | 97.5 | 0.26 | No good |
| I-5 | 0.38 | 98.4 | 0.43 | No good |
| I-6 | 0.31 | 98.9 | 0.19 | Accept/Good |
| I-7 | 0.37 | 98.5 | 0.15 | Accept |
| I-8 | 0.43 | 97.9 | 0.11 | No good |
| I-9 | 0.41 | 98.2 | 0.20 | No good |

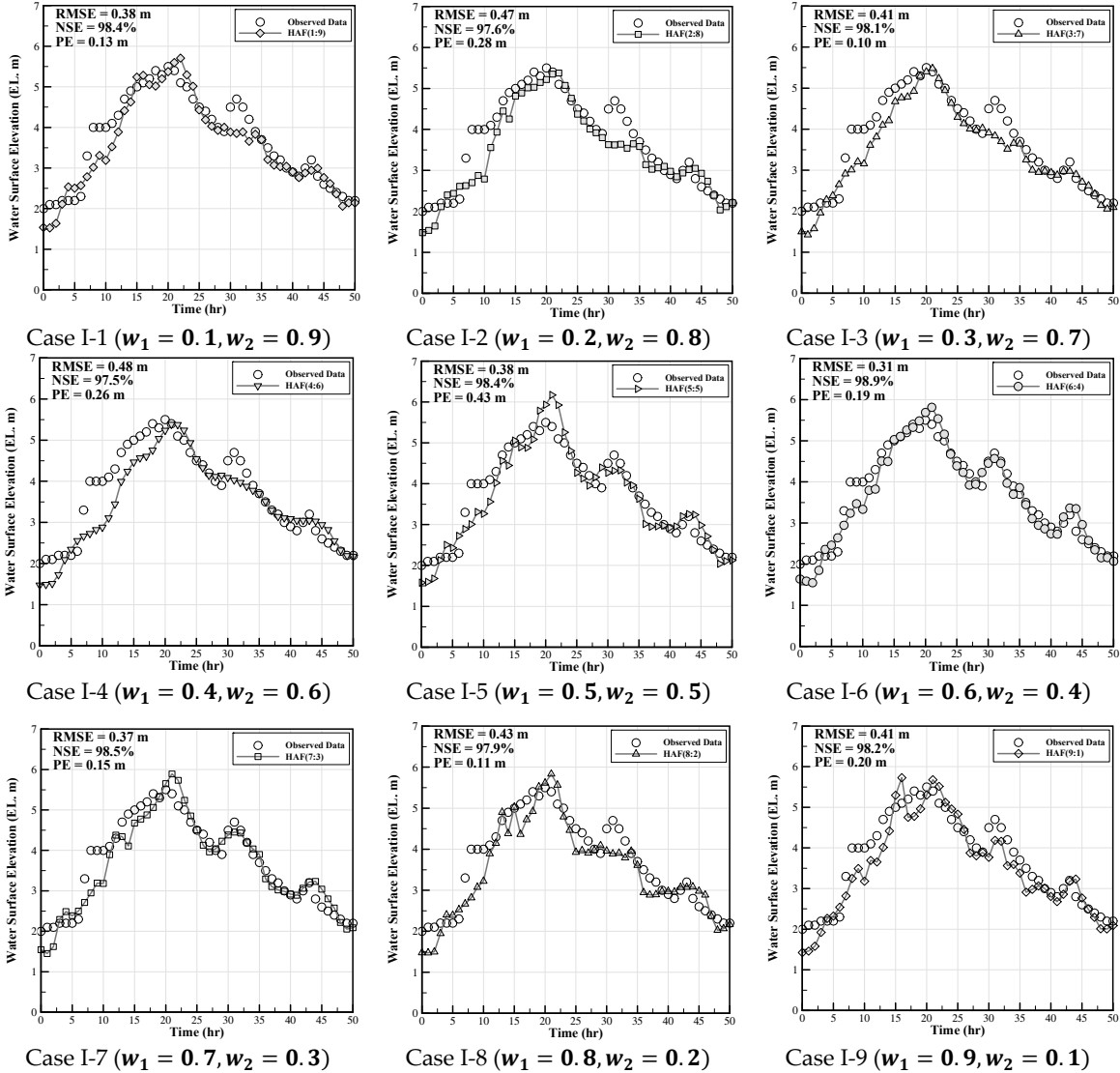

**Figure 5.** Results of the flood forecasting model in Scenario I.

The forecasting results were changed according to the weights (see Figure 6). And the forecasting results were well matched with the observed water surface elevation at $w_1 = 0.6$ and $w_2 = 0.4$. In terms of forecasting accuracy, the NSE was 98.9%, the RMSE was 0.31 m, and the PE was 0.19 m at $w_1 = 0.6$ and $w_2 = 0.4$, according to Table 7. The forecasting results have an uncertainty. Therefore, the confidence interval was examined by changing the weights (from $w_i = 0.1$ to $w_i = 0.9$ where, $i = 1,2$). Most of the forecasting results were under-forecasted and they were overfitted at high water surface elevations above the specified elevation (EL. 3.9 m) for flood management when compared with the observed data. The results of the confidence interval when using the hybrid activation function proposed in this study are summarized in Table 8.

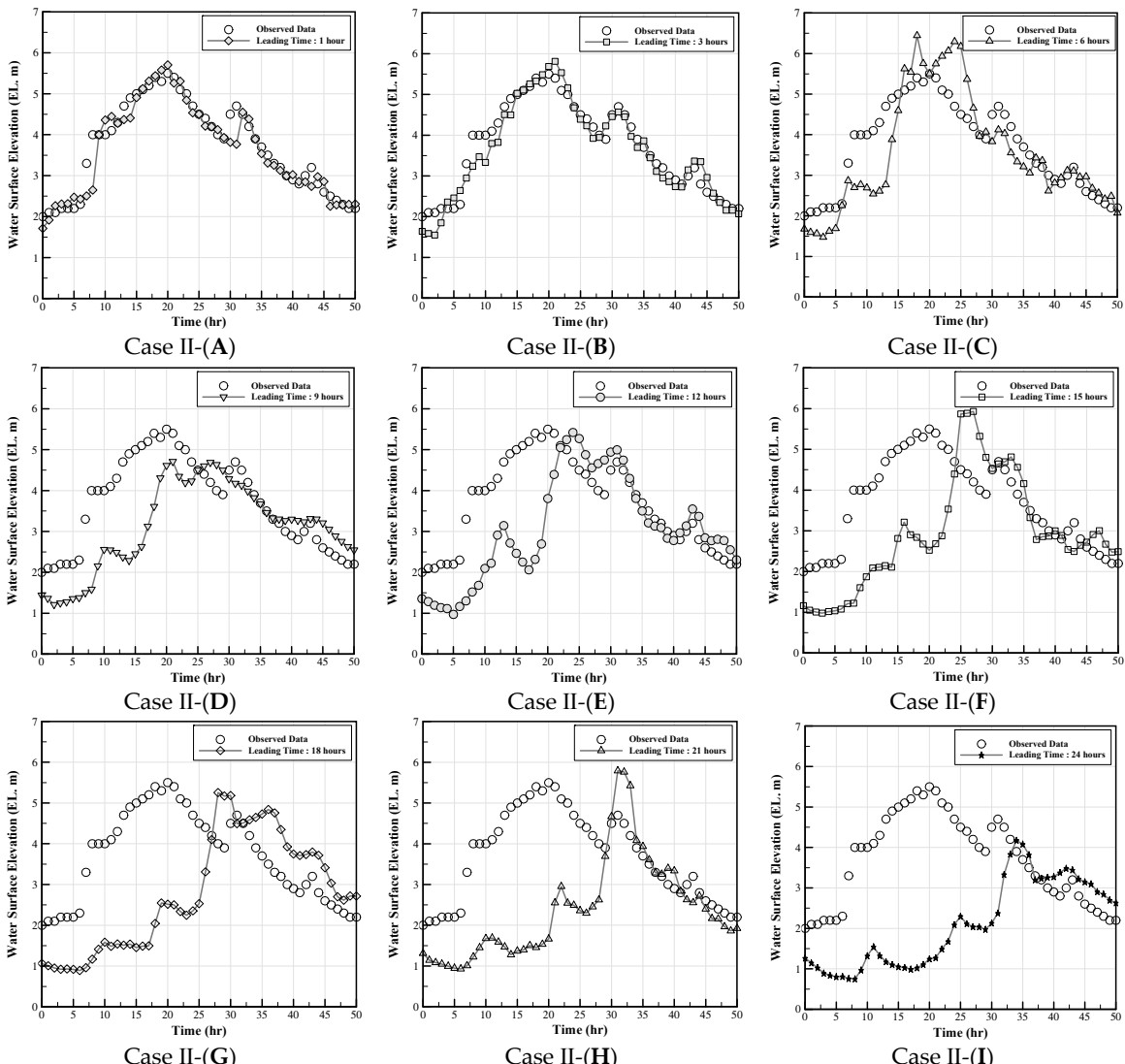

**Figure 6.** Results of the flood forecasting model according to Scenario II ((**A**) = 1 h leading time; (**B**) = 3 h leading time; (**C**) = 6 h leading time; (**D**) = 9 h leading time; (**E**) = 12 h leading time; (**F**) = 15 h leading time; (**G**) = 18 h leading time; (**H**) = 21 h leading time; (**I**) = 24 h leading time).

**Table 8.** Summary of confidence intervals for various weights in Scenario I.

| Case | CI * (%) | Accuracy ** (%) | Standard Deviation (m) | Remark |
|------|----------|-----------------|------------------------|--------|
| I-1 | 99.0 | 29.0 | 0.12 | No good |
| I-2 | 95.0 | 18.0 | 0.11 | No good |
| I-3 | 95.0 | 25.0 | 0.10 | No good |
| I-4 | 90.0 | 32.0 | 0.11 | No good |
| I-5 | 80.0 | 40.0 | 0.10 | No good |
| I-6 | 80.0 | 61.0 | 0.10 | Accept/Good |
| I-7 | 80.0 | 46.0 | 0.11 | Accept |
| I-8 | 95.0 | 32.0 | 0.11 | No good |
| I-9 | 99.0 | 32.0 | 0.14 | No good |

* CI = Confidence Interval; ** accuracy = total number of correct forecasting/total number of observed data (at 80.0% CI).

As shown in the I-6 case, the forecasting results of the hybrid activation function ($w_1 = 0.6$ and $w_2 = 0.4$) obtained the smallest confidence interval and a better performance than the other cases. In addition, the forecasting results of the hyperbolic tangent function and the rectified linear unit

function were compared with the forecasting results of the hybrid activation function ($w_1 = 0.6$ and $w_2 = 0.4$) proposed in this study at the high water surface elevation above the specified elevation for flood management (EL. 3.9 m) which caused the flood damage at the riverside. The results are summarized in Table 9.

**Table 9.** Comparison of the accuracy in various activation functions (above 3.9 m).

| Activation Function | RMSE (m) | NSE (%) | PE (m) |
|---|---|---|---|
| Hyperbolic Tangent ($w_1$= 0.0, $w_2$= 1.0) | 0.54 | −21.2 | 0.30 |
| Rectified Linear Unit ($w_1$= 1.0, $w_2$= 0.0) | 0.57 | −38.9 | 0.40 |
| Hybrid ($w_1$= 0.6, $w_2$= 0.4) | 0.32 | 58.1 | 0.19 |

When using the hybrid activation function, the RMSE decreased by 0.22 m compared with the hyperbolic tangent function and decreased by 0.25 m compared with the rectified linear function and the NSE increased by 79.3% compared with the hyperbolic tangent function and increased by 97.0% compared with the rectified linear function. Finally, the PE was 0.11 m and 0.21 m less than the hyperbolic tangent function and rectified linear unit function, respectively. From these results, the use of the hybrid activation function instead of the existing activation function had the effect of improving the forecasting accuracy at a high water surface elevation. However, it can be confirmed that the forecasting accuracy did not improve in all the data. Therefore, the hybrid activation function was more effective only for forecasting a high water surface elevation than the existing activation function because the forecasting water surface elevation in this study was highly affected by seasonal characteristics in Korea. Moreover, the time series data of water surface elevation, tidal level, precipitation and discharge have characteristics such as trend and seasonality. Even though the LSTM was generally used to forecast the time series data [22], the high water surface elevation was under-forecasted because of the lack of samples at a high water surface elevation. In particular, the Hangang River has a large coefficient of river regime in terms of river discharge. In addition, the original activation functions have the problems of a vanishing gradient and a dying neuron that occurred in the backpropagation training. Thus, the original activation functions have the limitation of needing to resolve the under-forecasted problem. Therefore, the hybrid activation function proposed in this study to partially solve the problems of the vanishing gradient and the dying neuron show the more accurate forecasting than the original activation functions, as shown in Table 9. Therefore, we conclude that the hybrid activation function was suitable for forecasting a high water surface elevation in Hangang River.

*6.2. Results from Scenario II*

The maximum forecasting time for the hybrid activation function applying the results of Scenario I ($w_1$= 0.6 and $w_2$= 0.4) is summarized in Table 10. In addition, the forecasting results and the observed data were compared for about 50 h from 5 July 2016 as the leading time was changed (see Figure 6).

**Table 10.** Summary of the accuracy for various leading times in Scenario II (Period: 5–7 July 2016 about 50 h, $w_1 = 0.6$ & $w_2 = 0.4$).

| Case | RMSE (m) | NSE (%) | PE (m) | Remark |
|------|----------|---------|--------|--------|
| II-(A) | 0.33 | 98.8 | 0.20 | Accept/Good |
| II-(B) | 0.31 | 98.9 | 0.19 | Accept/Good |
| II-(C) | 0.77 | 93.6 | 0.00 | Accept |
| II-(D) | 1.13 | 86.1 | 0.89 | Not Good |
| II-(E) | 1.31 | 81.2 | 1.69 | Not Good |
| II-(F) | 1.52 | 74.8 | 2.97 | Not Good |
| II-(G) | 1.93 | 59.3 | 2.98 | Not Good |
| II-(H) | 2.05 | 53.9 | 3.84 | Not Good |
| II-(I) | 2.33 | 40.8 | 4.25 | Not Good |

As the leading time interval increased, the value of water surface elevation was shifted to delay the peak event and under-forecasted the values in the rising rim. The RMSE increased sharply after 3 h of leading time while the NSE decreased sharply after 3 h of leading time (see Figure 7). Therefore, the good forecasting of the water surface elevation was up to 3 h in terms of RMSE and NSE. When using the hybrid activation function, the RMSE decreased by 0.10 m to 0.25 m compared with the hyperbolic tangent function and the rectified linear function, and the NSE increased by 3.2% to 20.1% compared with the hyperbolic tangent function and the rectified linear function. Finally, the PE was 0.03 m to 0.41 m less than the hyperbolic tangent function and the rectified linear unit function, respectively. From the results of Scenario II (see Table 11), it can also be considered that the use of the hybrid activation function could partially solve the vanishing gradient and dying neuron problems through accurate forecasting results at a high water surface elevation. Moreover, it was considered that the leading time interval was two times longer than the single activation functions when using the hybrid activation function. However, when using the hybrid function, it was shown that the accuracy of the forecasting was poor after 3 h of leading time interval. The reason for this was that the value of partial the autocorrelation coefficient of the water surface elevation at Hangang River was under the 95% confidence interval after 3 h of leading time interval (the range of 95% confidence interval was −0.2 to 0.2 in the partial autocorrelation analysis). Therefore, it was shown that the correlation of the water surface elevation was significant up to 3 h in Hangang River (see the small figure in Figure 8). However, in the case of PE, accurate forecasting was performed up to 6 h of leading time. Thus, the leading time interval was acceptable up to 6 h because the accurate forecasting of the peak water surface elevation is also important in flood forecasting.

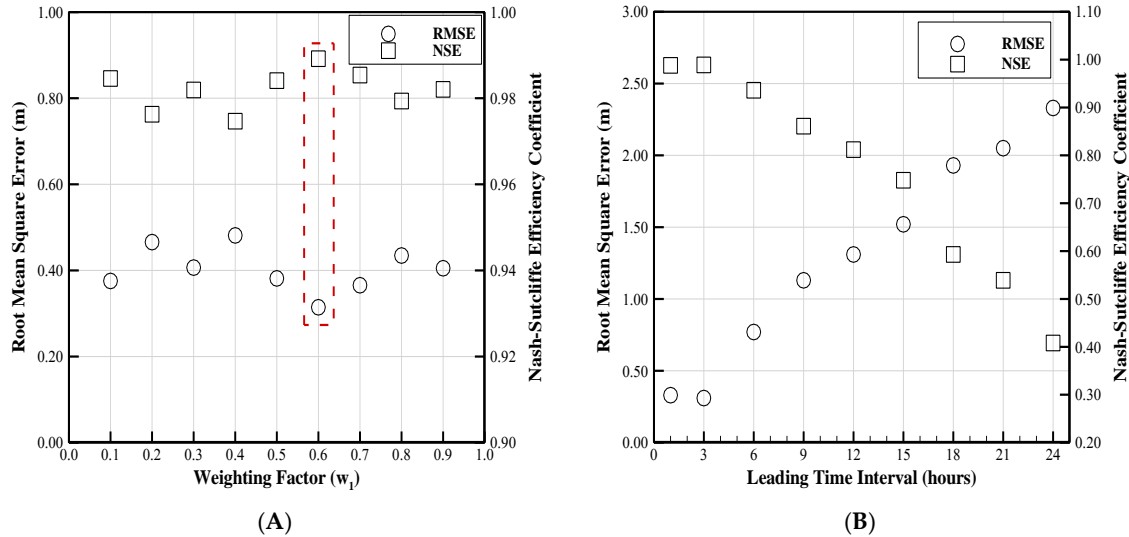

**Figure 7.** Overall results from Scenarios I and II. (**A**) Scenario I; (**B**) Scenario II.

**Table 11.** Comparison of the accuracy in various activation function in Scenario II.

| Case | Hyperbolic Tangent | | | Rectified Linear Unit | | | Hybrid | | |
|---|---|---|---|---|---|---|---|---|---|
| | RMSE (m) | NSE (%) | PE (m) | RMSE (m) | NSE (%) | PE (m) | RMSE (m) | NSE (%) | PE (m) |
| II-(A) | 0.43 | 95.2 | 0.25 | 0.42 | 95.6 | 0.23 | 0.33 | 98.8 | 0.20 |
| II-(B) | 0.45 | 92.3 | 0.25 | 0.48 | 93.4 | 0.21 | 0.31 | 98.9 | 0.19 |
| II-(C) | 0.90 | 86.8 | 0.49 | 0.88 | 88.6 | 0.41 | 0.77 | 93.6 | 0.00 |
| II-(D) | 1.23 | 80.8 | 1.05 | 1.28 | 83.3 | 0.95 | 1.13 | 86.1 | 0.89 |
| II-(E) | 1.55 | 78.2 | 1.80 | 1.48 | 79.5 | 1.90 | 1.31 | 81.2 | 1.69 |
| II-(F) | 1.86 | 66.2 | 3.11 | 1.82 | 68.3 | 3.02 | 1.52 | 74.8 | 2.97 |
| II-(G) | 2.08 | 48.3 | 3.55 | 2.03 | 52.1 | 3.23 | 1.93 | 59.3 | 2.98 |
| II-(H) | 2.33 | 38.6 | 4.12 | 2.28 | 40.9 | 4.03 | 2.05 | 53.9 | 3.84 |
| II-(I) | 2.58 | 20.7 | 4.65 | 2.66 | 25.8 | 4.55 | 2.33 | 40.8 | 4.25 |

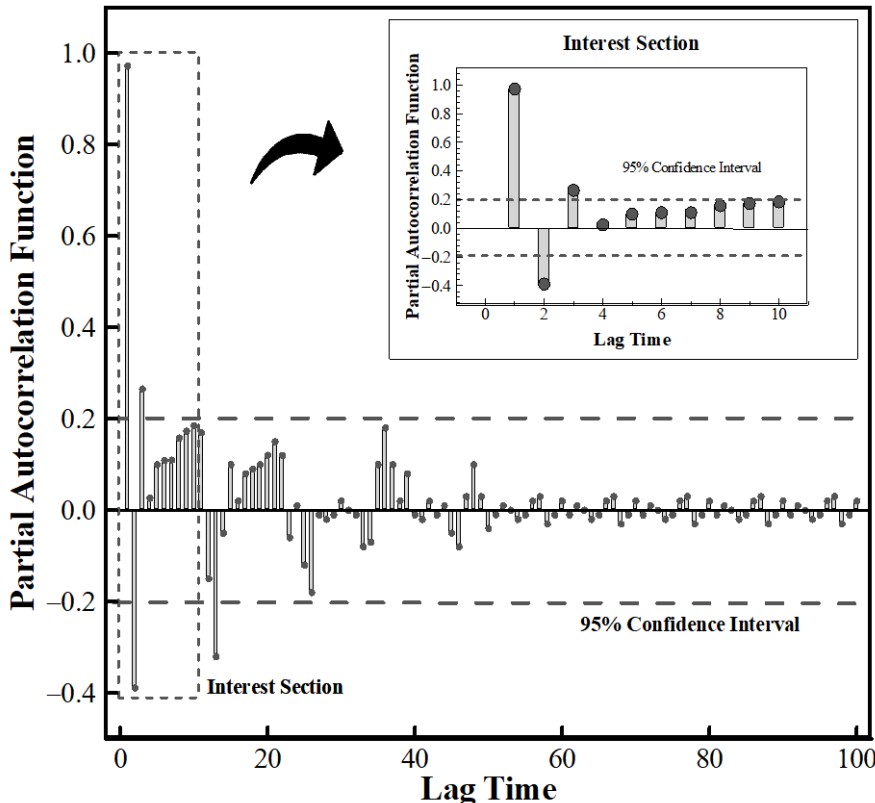

**Figure 8.** Partial autocorrelation of water surface elevation at Hangang River.

### 6.3. Flood Risk Assessment

The virtual flood forecasting, defined as the forecasting of flood occurrence with the data-driven model developed in this study, was performed using the model's results and the metrices for classification evaluation were used to evaluate the model. The criteria for flood forecasting was assumed to be above the specified elevation in Hangang River (EL. 3.9 m) and reviewed for one year, 2018. The metrices for the classification evaluation were accuracy, precision, and recall. Moreover, the confusion matrix for the binary classification and the corresponding array representation were used as shown in Table 12.

**Table 12.** Confusion matrix for binary classification and the corresponding array representation used in this study.

| | Actual Flood Class | Actual Non-Flood Class |
|---|---|---|
| Forecasting Flood Class | True Flood ($tf$) | False Non-Flood ($fn$) |
| Forecasting Non-Flood Class | False Flood ($ff$) | True Non-Flood ($tn$) |

The results of the classification evaluation are summarized in the Table 13 according to the activation function. In terms of the accuracy, all the activation functions have the same performance in flood forecasting. Moreover, in terms of the precision, the use of the hybrid activation function showed better performance than the existing activation functions and in terms of the recall, it was also shown that the hybrid function has a good performance for forecasting flood. However, the improvement of model performance with the hybrid activation function was not noticeable compared with the forecasting surface elevation with a hybrid activation function because the total samples of flood events were too small to evaluate the performance of model, especially the number of false non-flood conditions ($fn$) and false flood conditions ($ff$). Nevertheless, the model with the hybrid activation function was suitable for forecasting flood from the results of virtual flood forecasting because the accurate forecasting performed at a high water surface elevation with the hybrid activation function. In addition, it was considered that the accurate forecasting water surface elevation in the high water surface elevation conditions was an important factor in performing accurate flood forecasting.

**Table 13.** The results of the classification evaluation according to the activation function.

| Activation Function | Accuracy * | Precision * | Recall * |
|---|---|---|---|
| Hyperbolic Tangent ** ($w_1 = 0.0$, $w_2 = 1.0$) | 0.99 | 0.88 | 0.84 |
| Rectified Linear Unit *** ($w_1 = 1.0$, $w_2 = 0.0$) | 0.99 | 0.90 | 0.76 |
| Hybrid **** ($w_1 = 0.6$, $w_2 = 0.4$) | 0.99 | 0.95 | 0.84 |

* Accuracy = $(tf + tn)/(tf + tn + ff + fn)$; Precision = $(tf)/(tf + ff)$; Recall = $(tf)/(tf + fn)$; ** Hyperbolic Tangent ($tf = 19$, $tn = 8245$, $ff = 2$, $fn = 6$); *** Rectified Linear Unit ($tf = 21$, $tn = 8244$, $ff = 3$, $fn = 4$); **** Hybrid ($tf = 21$, $tn = 8246$, $ff = 1$, $fn = 4$).

## 7. Conclusions

The data-driven model was used for forecasting the water surface elevation in a tidal river and applied to Hangang River, Korea, in this study. The LSTM was constructed using the TensorFlow, an open-source library of deep learning and hourly data, such as precipitation, outlet discharge and tidal level were used as the input dataset through a *t*-test analysis. In particular, the hybrid activation function was proposed to resolve a specific issue of the single activation function, which is that they tended to under-forecast the results at the conditions for high water surface elevations above EL. 3.9 m, the designated elevation for flood management in Hangang River. The parameters of LSTM were determined through the sensitivity analysis, in which the number of hidden layers was 10, the learning rate was 0.005, and there were 1000 iterations. Moreover, sequence length, which is the most important parameter that determines the temporal amount of learning information during the learning time, was simulated for various leading times. Finally, the forecasting results, compared with the observed data, show improvement in the prediction accuracy (Scenario I) and the enhancement of the application range of the leading time interval (Scenario II) with the hybrid activation function.

First, for the application of the hybrid activation function, the optimal performance of the forecasting model was obtained when $w_1 = 0.6$ and $w_2 = 0.4$. The forecasting accuracy of all the data was presented to be 0.31 m in RMSE and 98.9% in NSE. However, in this case, the accuracy of the forecasting results was the same as the existing activation function, such as the hyperbolic tangent and ReLU. On the other hand, the RMSE was 0.32 m, NSE was 58.1%, and PE was 0.19 m at the conditions for high water surface elevation, which can cause flood damage at the riverside. These results show that the RMSE decreased by 0.22–0.25 m, PE also decreased by 0.11–0.21 m, and NSE increased up to 79.3%–97.0% compared with the existing single activation function. Therefore, the hybrid activation function proposed in this study was suitable for forecasting high-water levels above the specified elevation of Hangang River.

Secondly, the limitation of the application range of the leading time interval was obtained when the hybrid activation function was applied, which can be accurately forecasted at a high water surface elevation. We found a lower accuracy of forecasting at the longer leading time interval. The accuracy of forecasting was acceptable up to 3 h in terms of RMSE and NSE because the value of the partial autocorrelation coefficient about the water surface elevation at Hangang River was under the 95% confidence interval after the 3 h leading time interval. Therefore, it was considered that the correlation of the water surface elevation was significant up to 3 h in Hangang River. However, in the case of PE, the accurate forecasting was performed up to 6 h of leading time. Therefore, the leading time interval was acceptable up to 6 h because the accurate forecasting of the peak water surface elevation was also important in terms of flood forecasting.

Thirdly, the improvement of model performance for virtual flood forecasting with the hybrid activation function was not noticeable compared with the forecasting surface elevation with the hybrid activation function because the total samples of flood events were too small to evaluate the performance of the model. Nevertheless, the model using the hybrid activation function proposed in this study showed a better performance accuracy, precision, and recall than the other single activation functions when performing the virtual flood forecasting using the model's results of the water surface elevation (in which accuracy was 0.99, precision was 0.95, and recall was 0.84). From these results, it can be seen that the accurate forecasting of high-water surface elevations was the most important factor in performing more accurate flood forecasting.

In this study, the LSTM based on deep learning was used as a complementary means of a physical and numerical model for forecasting water surface elevation in a tidal river, Hangang River, Korea. In addition, the hybrid activation function was proposed to improve the accuracy of forecasting the high-water levels. In the near future, the forecasting results of the model proposed in this study will more accurately identify the effects of climate change on riverside and they are also expected to be used as a basis for establishing an emergency action plan (EAP) along riversides. In addition, if the information is provided to citizens through personal internet media, such as SNS, promptly, it will be possible to evacuate in advance and reduce human injury and damage.

**Author Contributions:** The following statements should be used "Conceptualization, H.J.Y. and S.O.L.; methodology, S.O.L.; software, H.J.Y.; validation, D.H.K., H.-H.K. and S.O.L.; formal analysis, H.J.Y.; investigation, D.H.K.; resources, H.J.Y.; data curation, H.J.Y.; writing—original draft preparation, H.J.Y. and S.O.L.; writing—review and editing, H.J.Y., H.-H.K. and S.O.L.; visualization, H.J.Y.; supervision, S.O.L.; project administration, S.O.L.; funding acquisition, S.O.L. All authors have read and agree to the published version of the manuscript.

**Funding:** This research was funded by Korea Environment Industry & Technology Institute (KEITI) through Water Management Research Program, gran number 12572.

**Acknowledgments:** This work was supported by Korea Environment Industry & Technology Institute (KEITI) through Water Management Research Program, funded by Korea Ministry of Environment (127572).

**Conflicts of Interest:** The authors declare no conflict of interest.

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
