# Peer review of "Data Driven Water Surface Elevation Forecasting Model with Hybrid Activation Function—A Case Study for Hangang River, South Korea"

_applsci, doi:10.3390/app10041424_

Round 1

Reviewer 1 Report

The authors of this article need to show the confidence intervals of their predictions explicitly. This will depend upon the confidence intervals arising in the estimation of their hybrid model from data.

The danger with a neural network approach is that overparametrization can result in the blow up of confidence intervals. How do you show that this does not happen in this case? That the interpolation is safe for flood prediction?

Author Response

Response to Reviewer 1 Comments

Point 1:The authors of this article need to show the confidence intervals of their predictions explicitly. This will depend upon the confidence intervals arising in the estimation of their hybrid model from data

Response 1: Thank you for your comments. We examined the confidence interval of forecasting results. And the results of confidence interval with hybrid activation function was presented in table. (see line 445 - 456, highlighted section in revision manuscripts)

Point 2:The danger with a neural network approach is that overparameterization can result in the blow up of confidence intervals. How do you show that this does not happen in this case? That the interpolation is safe for flood prediction?

Response 2: Thank you for your comments. 

First, we find the previous research, And, they concluded that the combination of precipitation, outlet discharge of dam, water surface elevation and tidal level was good performance to forecast water surface elevation in Hangang river. So, we think the problems of overparameterization don’t occurred in this study. (see line 298-303, highlighted section in revision manuscripts)

Second, the interpolation is the common correction method about outlier and missing value of hydrology data. And the most outliers and missing values of hydrology data were occurred in the non-flood season which is outside of the scope in this study. Therefore, we think that the interpolation is safe for flood forecasting. And the forecasting results at high water surface elevation above the specified elevation for flood management (EL. 3.9 m) were over-forecasted. So, it showed the conservative results when assessing flood risk (see line 330 - 331, highlighted section in revision manuscripts).

We attached the file to revision manuscripts.

Thank you.

Reviewer 2 Report

This paper is well organized and rational in structure. Despite the good use of the English language, the exposition of the subject is very basic, but at the same time follows a logical thread. The following suggestions can be addressed such as:

Move the detail literature studies from introduction to literature study section. Only put the summary of literature in the introduction. Put as much as detail information regarding the experimental environment such as python, numpy, pandas, tensorflow version as well as the computer specification used for the experiment. Add more literature related to the activation function used in the study. Why the hybrid outperformed the original activation function? 

Author Response

Response to Reviewer 1 Comments

Point 1:First, move the detail literature studies from introduction to literature study section. Second, Only put the summary of literature in the introduction

Response 1: Thank you for your comments.

First, we put the summary of literature in the introduction (see line 36 - 78, highlighted section in revision manuscripts).

Second, we move the detail literature studies from introduction to literature study section. (see line 81 - 127, highlighted section in revision manuscripts)

Point 2:Put as much as detail information regarding the experimental environment such as python, numpy, pandas, tensorflow version as well as the computer specification used for the experiment

Response 2: Thank you for your comments.

We add a detail information regarding the experimental environment. (see line 362 - 370, highlighted section in revision manuscripts )

Point 3: First, add more literature related to the activation function used in the study. Second, why the hybrid outperformed the original activation function?

Response 3: Thank you for your comments.

First, we add more literature related to the activation functions such as sigmoid, hyperbolic tangent and rectified linear unit. (see line 191 - 225, highlighted section in revision manuscripts )

Second, we add the reason why the hybrid activation function was outperformed than the original activation function. (see line 477 - 487, highlighted section in revision manuscripts )

We attached the file to revision manuscripts.

Thank you.

Reviewer 3 Report

Dear Authors,

Thank you so much for providing this interesting article. The issue encountered in this article is of great importance. As far as I observed the paper is written in a very professional manner. Below are my comments,

1: Background and previous literature review are soundly provided at the beginning of the article but as they are dealing with two basic issues i.e., optimization hurdle and vanishing gradient in RNN, I think they should have explained these problems in few words in the proposed problem. 

2: On line 180 they mention the change in weights. What criteria the weights are following to change and what is the corresponding effect on performance? How do they affect the whole system?

3: From equation 9 to 11 the subscript "i" is not mentioned in the main text.  

3: I found many grammatical mistakes such as on line 39 they used "and" line 76and 80 is poorly written. There is a full stop followed by "and" on line 96 and many more English mistakes are found in this article. 

The proposed hybrid technique is good and the performance is also quite nice. 

Best of luck! 

Author Response

Response to Reviewer 1 Comments

Point 1:Background and previous literature review are soundly provided the beginning of the article but as they are dealing with two basic issues i.e., optimization hurdle and vanishing gradient in RNN, I think they should have explained these problems in few words in the proposed problem

Response 1: Thank you for your comments.We explained the problems of optimization hurdle and vanishing gradient in few words (see line 132 - 136, highlighted section in revision manuscripts )

Point 2: On line 180 they mention the change in weights. What criteria the weights are following to change and what is the corresponding effect on performance? How do they affect the whole system

Response 2: Thank you for your comments.

First, the hybrid activation function is composed of hyperbolic tangent function and rectified linear unit function to solve the problems of general activation functions such as vanishing gradient problem, dying neuron problem and so on. So, the weights changes to check the variation of forecasting accuracy. (see line 405 - 411, highlighted section in revision manuscripts)

Second, the effect of improving the forecasting accuracy was confirmed by applying the hybrid activation function compared with the existing activation function. The results are summarized in Table 9. (see line 472 - 474, highlighted section in revision manuscripts)

Point 3: From equation 9 to 11 the subscript “i" is not mentioned in the main text

Response 3: Thank you for your comments.

We mentioned the subscript “i" from equation 9 to 11 in the main text (see line 251 -252, highlighted section in revision manuscripts)

Point 4: I found many grammatical mistakes such as on line 39 they used “and” line 76 and 80 is poorly written. There is a full stop followed by “and” on line 96 and many more English mistakes are found in this article

Response 4: Thank you for your comments.

We revised the grammatical mistakes.

For forecasting the flooding occurred in riverside, many countries have provided flood forecasting system services such as Advanced Hydrologic Prediction Service (AHPS in United States), Evaluation et Suivi des Pluies en Agglom ration pour Devancer I’Alerte (ESPADA in France) by using hydrological data (see line 38-39, highlighted section in revision manuscripts)

Chen et al. [17] constructed and ANN-based prediction model in river and reviewed the applicability of model. (see line 100, highlighted section in revision manuscripts) → delete the “and”

So, the recent research trend about forecasting water surface elevation using artificial neural network was changed from ANN, recurrent neural network (RNN) models to long short-term memory (LSTM) model. (see line 104-106, highlighted section in revision manuscripts)

Jung et al. [24] used the LSTM model to forecast the upstream water surface elevation in the Geumgang river basin. The accurate forecasting was performed for the entire water surface elevation. However, the forecasting result was underestimated at high water surface elevation. (see line 119, highlighted section in revision manuscripts)

We attached the file to revision manuscripts.

Thank you.

Round 2

Reviewer 1 Report

The authors have addressed all my concerns.